# Maternal stress and placental function; *ex vivo* placental perfusion studying cortisol, cortisone, tryptophan and serotonin

**Line Mathiesen**[1]*, **Cecilie Bay-Richter**[2], **Gregers Wegener**[2], **Nico Liebenberg**[2], **Lisbeth E. Knudsen**[1]

**1** Department of Public Health, Section of Environmental Health, University of Copenhagen, Copenhagen, Denmark, **2** Department of Clinical Medicine, Translational Neuropsychiatry Unit, University of Aarhus, Aarhus, Denmark

* lima@sund.ku.dk

## Abstract

### Background

Exposure to maternal stress during pregnancy can have adverse effects on the fetus, which has potential long-term effects on offspring´s development and health. We investigated the kinetics and metabolism of the hormones and amino acids: cortisol, cortisone, tryptophan and serotonin in the term placenta in an *ex vivo* human placental perfusion model. The placentas used in the experiments were donated from families participating in the Maternal Stress and Placental Function project with a known maternal stress background.

### Method

Cortisol, cortisone, tryptophan and serotonin were added simultaneously to the maternal side in the 6 hour *ex vivo* term human recirculating placental perfusion model, in four experimental set-ups: without inhibitors, with carbenoxolone -that inhibits cortisol metabolism into cortisone, with fluoxetine that inhibits the serotonin transporter, and with PCPA that inhibits metabolism of tryptophan into serotonin. The concentration of cortisol and cortisone, and tryptophan and serotonin were quantified using UPLC and HPLC-MS respectively.

### Results

Cortisol was rapidly metabolized into cortisone in the placenta, to a somewhat lesser degree when adding the inhibitor carbenoxolone, resulting in higher fetal exposure to cortisol. Serotonin was also rapidly metabolized in the placenta. When adding fluoxetine a peak of fetal serotonin levels was seen in the first hour of the perfusion. No effect was seen of the maternal stress levels on placental transport kinetics in this study.

### Conclusion

Inhibiting the metabolism of cortisol in the placenta increased fetal exposure to cortisol as expected. Unexpectedly we saw an increased fetal exposure to serotonin when inhibiting the serotonin transporter, which may be related to the increased serotonin concentration on

**Data Availability Statement:** All relevant data are within the paper and its Supporting Information files.

**Funding:** The Maternal Stress and Placental Function project was funded by AFAAR/NEAVS (the American Fund for Alternatives to Animal Research/New England Anti-Vivisection Society Fellowship Grant for Alternatives to Animal Research in Women's Health and Sex Differences) (LM), and internal funding from the University of Copenhagen. HPLC used to quantify antipyrine in perfusion samples was funded by Alternativfondet (LM). Analysis of cortisol and cortisone in perfusion samples was funded by Læge Sofus Carl Emil Friis og Hustru Olga Doris Friis' Legat (LM). Analysis of serotonin, serotonin metabolites and tryptophan in perfusion samples was funded by internal funding from University of Aarhus (GW). The funders had no role in study design, data collection and analysis, decision to publish, or preparation of the manuscript.

**Competing interests:** GW reports to have received lecture/consultancy fees/research support from H. Lundbeck A/S, Servier SA, Astra Zeneca AB, Eli Lilly A/S, Sun Pharma Pty Ltd, Pfizer Inc, Shire A/S, HB Pharma A/S, Arla Foods A.m.b.A., Alkermes Inc, Mundipharma International Ltd., and Jannsen Pharmaceuticals A/S. No other conflicts of interests are reported. This does not alter our adherence to PLOS ONE policies on sharing data and materials.

**Abbreviations:** 5-HIAA, 5- hydroxyindoleacetic acid, a metabolite of serotonin; 5-HT, 5-hydroxytryptamine, also called serotonin; 11β-HSD2, 11β-Hydroxysteroid dehydrogenase, a placental enzyme that transforms cortisol into cortisone; BMI, body mass index; HSA, human serum albumin; DASS-42, a questionnaire measuring depression, anxiety and stress; IDO1 and 2, indoleamine 2,3-dioxygenase-1 and -2, an enzyme in the kynurenine pathway; MAO A, Monoamine oxidase A; NEO-FFI, NEO Five Factor Inventory, a measure of the personality traits of Neuroticism, Extraversion, Openness, Conscientiousness and Agreeableness; PRA, Pregnancy-Related Anxiety; SERT, Serotonin transporter; SSRI, Selective serotonin receptor inhibitor; TPH1 and 2, Tryptophan hydroxylase 1 and 2, an enzyme in the serotonin pathway.

the maternal side of the placenta. No effect on placental kinetics were evident on maternal stress levels during the pregnancy as the majority of participating mothers had normal stress levels.

## Introduction

The mechanisms of maternal psychosocial stress affecting the fetus during pregnancy are assumed to be regulated by placental transfer of hormones, through changes in the expression of placental receptors and enzymes (for reviews, see [1,2]).

The placenta consists of different cell layers in the human placenta and the layer most representative of the placental transport and metabolism is the syncytiotrophoblast. This cell layer expresses the enzyme 11β-Hydroxysteroid dehydrogenase (11β-HSD2), which transforms 80–90% of the maternal cortisol to cortisone before translocating on to the umbilical and fetal blood [3,4]. The activity of 11β-HSD2 has been linked to the effect of maternal psychosocial stress on the offspring, as it protects the fetus from the high cortisol plasma levels of the pregnant woman (for a review see [5]).

Serotonin (5-hydroxytryptamine, 5-HT) is a neurotransmitter that also plays a role during early development of the human fetus, where it acts as a growth factor, regulating the development of fetal neural systems. Serotonin is important for the development of the hypothalamic-pituitary-adrenal (HPA) axis, and acts as a mediator between early life experience and subsequent behaviour [6]. Transplacental transport of serotonin occurs via the 5-HT transporter (SERT) and via placental metabolism of tryptophan [7,8]. In the placenta serotonin is metabolized into 5- hydroxyindoleacetic acid (5-HIAA) by monoamine oxidase A (MAO A) [9].

Tryptophan is the essential amino acid utilized for the synthesis of serotonin and the hormone melatonin, although degradation of tryptophan occurs mostly (95%) along the kynurenine pathway with one of the end products being nicotinamide adenine dinucleotide (NAD$^+$). The first step in the kynurenine pathway occurs via three enzymes, indoleamine 2,3-dioxygenase-1 and -2 (IDO1 and IDO2), and tryptophan 2,3-dioxygenase (TDO) which can be induced by cortisol. Tryptophan hydroxylases (TPH1 and TPH2) is part of the pathway to synthesize serotonin. It is suggested that both IDO1, IDO2, and TDO and TPH are located in the human placenta, where they play a part in utero-placental immune system regulation, feto-maternal tolerance, antioxidant capacity and vasorelaxation [10].

In an earlier study from our group, the effect of prenatal maternal psychosocial stress on the adjusted fetal cortisol exposure (AFCE) was described in a normal pregnant population [11]. The AFCE was calculated to represent the relative amount of cortisone produced by the placenta measured in fetal blood in relation to how much cortisol crosses the placenta unmetabolized from maternal blood, corrected for the interindividual differences of these hormone levels in the population–i.e. the placental exposure. The AFCE provide a measure of the activity of the enzyme 11β-HSD2 and thus an increase in the AFCE represents a relative increase in fetal cortisol exposure.

Direct characterization of transport and metabolism of stress-related hormones in the human placenta is of significant importance to understand potential pathophysiological mechanisms. Therefore, using placentas from a previously studied population [11] and *ex vivo* placental perfusion, the aims of the present work was in human term placentas with known prenatal stress levels to describe the transport and metabolism of i) cortisol, ii) cortisone, iii) tryptophan and iv) serotonin. Finally, addition of inhibitors of placental enzymes 11β-HSD2,

SERT and THP aimed to v) give insight into the supposed effect of maternal stress on the placenta and thus fetal exposure.

## Materials and methods

The current study is a part of the '*Maternal Stress and Placental Function'* project, conducted in Copenhagen, Denmark, described in detail in a previous publication [11]. Briefly, participants were pregnant women giving birth at Copenhagen University Hospital. The project was approved by the Regional Scientific Ethical Committee of Copenhagen (H-15006254) and the Danish Data Protection Agency (2015-41-4208). All women were informed about the aim of the study, gave written informed consent and were instructed to answer four written questionnaires in order to measure: 1: relevant personal factors such as socioeconomic status, use of medication, smoking, and alcohol consumption during and before pregnancy, 2: pregnancy-related anxiety, 3: personality, and 4: prevalence of prenatal depression, anxiety and stress. Twenty-two placentas were set up in placental perfusion. Of these, 2 perfusions had a leak of medium from fetal circulation and were unsuccessful,5 perfusions showed a leak of medium from the fetal circulation after 4 hours perfusion and were thus unsuccessful after this time-point, and 15 were successful 6-hour perfusions.

### Psychometric measures

Maternal *state* stress was defined as the degree of depression, anxiety and stress (DASS) experienced during the pregnancy, pregnancy- and birth-related thoughts and anxiety (PRA), and the experience of major life events during pregnancy. Maternal *trait* stress was represented by the maternal personality traits: neuroticism and conscientiousness, using the Danish version of the NEO-FFI inventory. These measures were assessed as described in a previous paper [11].

### Placental perfusion

The basic placental perfusion method was performed in Copenhagen, and is previously described and validated [12,13]. In this study, the perfusions were performed on placentas delivered by women participating in the '*Maternal Stress and Placental Function study'*, also donating blood and umbilical cord blood. To set up the perfusion, the placenta was infused with Krebs ringer buffer supplemented with heparin directly after birth, and cord blood sampling at the Copenhagen University Hospital. Hereafter, it was transported to the perfusion laboratory were a suitable artery-vein pair was cannulated to perfuse a single cotyledon. The cotyledon was removed from the surrounding tissue and placed in the perfusion chamber maternal side up. The perfusion chamber with the placental lobule was then moved to a heated flowbench (37˚C) where fetal artery cannula was attached to a peristaltic pump. If the flow through the fetal vessels indicated that they were intact, the maternal side was bluntly cannulated with three metal cannula attached to a tube connected to a peristaltic pump, perfusing the maternal side of the cotyledon with 300 ml RPMI 1640 cell culture medium to wash out maternal residual blood. After this washing step, the fetal and maternal systems were closed to recirculate perfusion medium (100 ml RPMI 1640 cell culture medium with physiological levels of human serum albumin (HSA)) in a 1 hour pre-perfusion, the maternal outflow was drained by gravity (Fig 1).

Following the pre-perfusion was the six hour perfusion with 100 ml fresh perfusion medium in the maternal circulation added test substances (All from Sigma-Aldrich, Broendby, Denmark): physiological levels of cortisone (0.2 µg/ml), cortisol (0.2 µg/ml), tryptophan (13 µg/ml) and serotonin (0.3 µg/ml) (n = 22). Five of these perfusions were added

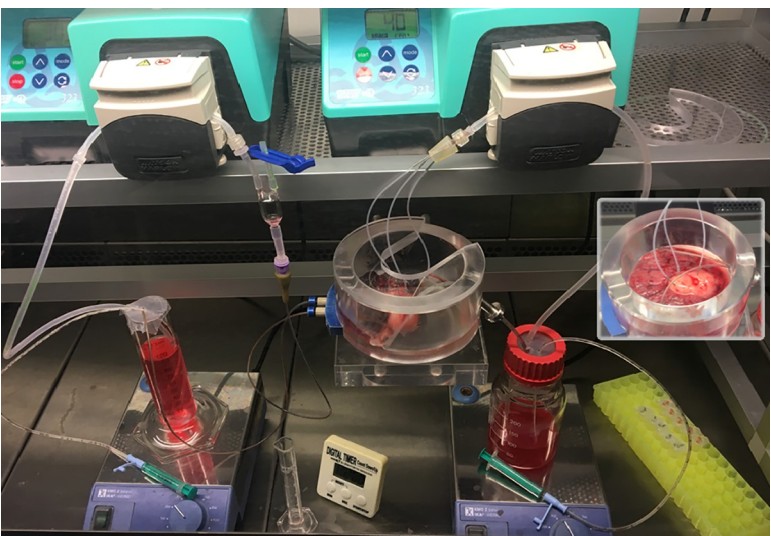

**Fig 1. The placental perfusion system set-up.** The full picture shows the perfusion set-up in the heated flowbench. At the left is the fetal reservoir and tubes with bubble-trap recirculating perfusion medium. In the middle is the placental lobule in the perfusion chamber. At the right is the maternal reservoir and tubes, recirculating medium, with three maternal cannula and gravitational collection of the outflow. The small cutout picture at the right is a closer view of the lobule and the three maternal cannula, with the light colored cotyledon caused by absence of blood.

carbenoxolone (50 μg/ml), an inhibitor of 11β-HSD2, four were added fluoxetine (SERT inhibitor) (0.5μg/ml), and five were added PCPA (binds irreversibly to tryptophan hydroxylase) (300μg/ml) (All from Sigma-Aldrich, Broendby, Denmark). The test substances and the control substance antipyrine were added to the maternal reservoir at start perfusion (time 0). Samples were taken out from maternal and fetal reservoir at time 0, 2, 30 minutes and thereafter at 30 minutes intervals until the end of perfusion. Oxygen level, glucose levels and pH in maternal and fetal circulation and fetal outflow, were measured during the perfusion using an ABLflex 90 blood gas analyzer (Radiometer, Denmark), and where regulated to stay in physiological levels. The volume of the fetal reservoir was observed after each sample, and at the end of perfusion, fetal and maternal volumes were measured. Success criteria were fetal system volume loss of less than 3 ml/h and antipyrine feto-maternal ratio (FM-ratio) above 0.75 after 150 minutes.

## Quantification of antipyrine

Supernatant (200 μL) was mixed with Acetonitrile (200μL, VWR, Fontenay-Sous-Bois, France) to precipitate protein, containing 10 μg/mL phenacetin (*p*-acetopenetidide 97%, Acros organics, Geel, Belgium) as internal standard. Antipyrine (98%, Aldrich-Chemie, Steinheim, Germany) and phenacetin was analyzed using a reverse phase LaChrom HPLC system equipped with a L-7100 pump, an L7200 autosampler, an D-7000 interface, a L-7300 column oven and a L-7400 UV detector (Merck, Hitachi). The stationary phase was a C18 column (NUCLEOSIL C-18, ODS, 20 mm × 4.6 mm, 5 μm particles) with a Security Guard precolumn (Phenomexes C-18, ODS, 4 mm *L* × 3.0 mm i.d.). Mobil phase was a degassed methanol (HPLC grade, Applichem, Biochemia, Damstadt, Germany) and water (45/55, v/v) solution adjusted to a flow rate of 1 mL/min. Injection volume was 25 μL, oven temperature 31˚C, and detection absorbance 254 nm.

## Quantification of cortisol and cortisone

The quantification method of cortisol and cortisone is described in detail elsewhere [11].

In brief: Sample analysis was performed using a Waters (Milford, MA, USA) Acquity UPLC system with a Kinetex® 2.6 μm EVO C18 column (100Å 100x2.1 mm; Phenomenex, Torrance, CA, USA). Column temperature was 50˚C, flow rate was 500 μl/min, and injection volume was 5 μl. The total analysis time was 9 minutes per sample. The mobile phase was a gradient of a mixture of an aqueous mobile phase 0.1% $NH_4OH$ (v/v) in water (mobile phase A) and an organic phase containing 0.1% $NH_4OH$ (v/v) in MeOH (mobile phase B).

## Quantification of tryptophan, serotonin and -metabolites

Tryptophan, serotonin (5-HT) and the 5-HT metabolite 5-hydroxyindoleaetic acid (5-HIAA) were examined by use of UHPLC. Perfusate samples were mixed at a ratio of 1:10 with 0.2 M $HClO_4$ and centrifuged at 14000 g for 10 min. The supernatant was transferred to Costar cellulose acetate filter tubes (0.22 μm; Corning Inc) and centrifuged at 4˚C for 10 min at 14 000 x g. For the measurement of 5-HT and 5-HIAA the filtrate was injected directly into the high-pressure liquid chromatography system. For the measurement of Trp 0.2 M perchloric acid was used to dilute the samples a further 50x. Chromatographic conditions were as follows: equipment consisted of Dionex Thermo Scientific Ultimate 3000 model isocratic pump and auto-sampler equipped with a Phenomenex Kinetix C18 2.6 μm, 150 x 4.6mm particle column. Detection of 5-HT and 5-HIAA was carried out using a Thermo Scientific Dionex model 6011RS ultra Coulometric Analytical cell (E1: -150 mV: E2: +250 mV vs Pd reference). The column was maintained at 27˚C while eluting the analytes with a MDTM mobile phase (Thermo Scientific Dionex Test Phase, 70–3829) at a flow rate of 1.0 mL/min. Tryptophan measurements were performed using the same conditions as for 5-HT and 5-HIAA but with cell potentials at E1: +250 mV: E2: +550 mV vs Pd reference.

## Statistical analysis

All statistical testing was performed in IBM SPSS Statistics 24. Correlations were tested using bivariate correlations test. Descriptive results are presented as mean and range. Due to small group sizes, effects of inhibitors on the transplacental transfer of cortisol, cortisone, tryptophan and serotonin is not tested statistically, and figures are depicted as mean without standard deviation. A p-value less than 0.05 is interpreted as statistically significant.

## Results

Out of all placentas donated and handled, 31% resulted in a successful 6-hour perfusion (see S1 Table).

Perfusion variables and substances added are shown in Table 1. All perfusions are simultaneously added cortisol, cortisone, tryptophan and serotonin. Out of the performed 22 perfusions, 7 fetal circulations became leaky (volume loss on fetal side > 3ml/hour) after 1–5.5 hours perfusion. Only samples from non-leaky parts of perfusions (volume loss in fetal system < 3ml/hour) until and over 4 hours are part of the graphs presented. All perfusion data are interpreted according to the mean of 3–8 repeated perfusions presented as percentage (%) of added substances in each individual perfusion, for raw data with concentrations (in nM) see S1 Data

The majority of the donating mothers were non-smokers who did not drink alcohol during the pregnancy. The age span was 24 years with a mean age of 35, and a mean body mass index (BMI) before pregnancy of 23. Almost half of the women had a chronic disease, 6 had

**Table 1. Details of each perfusion performed in the maternal stress and placental function project.** All perfusions (n = 22) are added cortisol (0.2 µg/ml), cortisone (0.2 µg/ml), tryptophan (13 µg/ml) and serotonin (0.3 µg/ml) on maternal side at time 0.

| ID | Antagonist | Time birth to lab (min) | Cotyledon weight (g) | Fetal flow (ml/2min) | Mean $O_2$ Maternal | Mean $O_2$ Fetal | Fetal volume loss 6h (ml/h) | Perfusion ok (h) |
|----|------------|------|------|-----|------|------|------|---|
| 1 | none | 33 | 20 | 5.9 | 15.2 | 13.1 | 2.0 | 6 |
| 2 | none | 40 | 29 | 6.0 | - | - | 1.4 | 6 |
| 3 | none | 33 | 10 | 5.8 | - | - | 0.1 | 6 |
| 4 | none | 36 | 20 | 5.8 | 17.2 | 9.4 | 0.9 | 6 |
| 5 | none | 44 | 13 | 6.2 | 11.2 | 7.0 | 1.4 | 6 |
| 6 | none | 57 | 20 | 6.6 | 14.6 | 5.8 | 2.6 | 6 |
| 7 | none | 50 | - | 5.6 | 13.6 | 10.2 | 0.4 | 6 |
| 8 | none | 62 | 11 | 5.4 | 16.5 | 12.7 | 0.6 | 6 |
| 9 | carbenoxolone | 33 | 13 | 6.0 | 15.0 | 9.4 | 1.1 | 6 |
| 10 | carbenoxolone | 39 | 18 | 6.7 | - | - | 10.1 | 1 |
| 11 | carbenoxolone | 40 | 24 | 6.1 | 12.3 | 5.4 | 2.7 | 6 |
| 12 | carbenoxolone | 74 | 11 | 6.1 | 12.6 | 6.0 | 0.4 | 6 |
| 13 | carbenoxolone | 40 | 7 | 6.0 | 16.0 | 6.4 | 8.6 | 3.5 |
| 14 | fluoxetine | 43 | 16 | 6.3 | 16.5 | 9.2 | 1.9 | 6 |
| 15 | fluoxetine | 32 | 20 | 6.0 | 9.9 | 4.1 | 5.0 | 5.5 |
| 16 | fluoxetine | 42 | 16 | 6.4 | 16.9 | 16.4 | 6.4 | 5 |
| 17 | fluoxetine | 31 | 22 | 6.1 | 15.5 | 7.0 | 4.4 | 5.5 |
| 18 | PCPA | 44 | 17 | 6.3 | 15.6 | 11.0 | 1.0 | 6 |
| 19 | PCPA | 50 | 7 | 5.9 | 14.7 | 2.26 | 9.6 | 4 |
| 20 | PCPA | 38 | 22 | 5.7 | 11.1 | 4.1 | 0.7 | 6 |
| 21 | PCPA | 42 | 30 | 5.7 | 12.7 | 2.5 | 8.4 | 4 |
| 22 | PCPA | 71 | 14 | 5.7 | 17.3 | 9.5 | 1.3 | 6 |

gestational complications and 7 took medication during pregnancy (analgesics, antibiotics, cobalamin, asthma medication and insulin) (see S2 Table).

## Cortisol and cortisone

The concentration of cortisol in the maternal circulation decreased rapidly to below 10% after 60 minutes perfusion with a plateau after approximately two hours perfusion. In the fetal

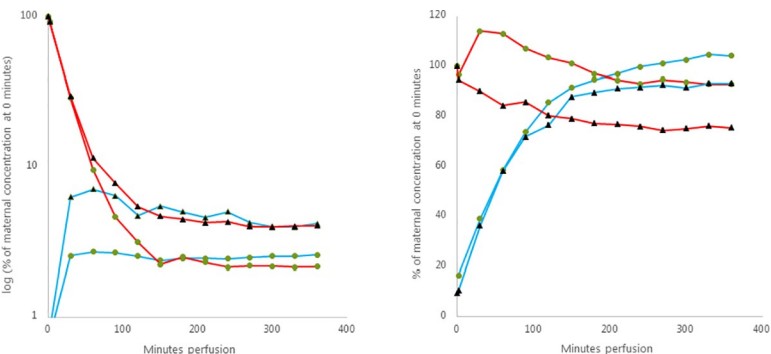

**Fig 2. Placental perfusion with cortisol and cortisone with and without carbenoxolone.** The graphs show maternal (red) and fetal (blue) mean concentration of cortisol (left graph) and cortisone (right graph), without (green circle, n = 8) and with (black triangle, n = 3) the addition of carbenoxolone. The concentration is presented as percentage (%) of the concentration found in the maternal sample at time 0 minutes perfusion.

circulation, the concentration of cortisol increased rapidly in the first thirty minutes and reached a plateau at equilibrium with the maternal side concentration after two hours perfusion. The addition of carbenoxolone caused a somewhat slower maternal decrease and higher fetal increase in cortisol concentration and a plateau around 4% of initial maternal concentration, compared to a plateau around 2% for the perfusions without carbenoxolone (Fig 2 left graph).

The perfusions with cortisone showed an increase in cortisone concentration in the maternal circulation followed by a slow decrease. There was an immediate rapid increase of cortisone in the fetal circulation that surpassed the concentration on maternal side within a few hours perfusion. When adding carbenoxolone the initial increase of cortisone in the maternal circulation failed to appear but the slow decrease that followed was similar. The initial increase in the cortisone concentration in fetal circulation was similar to perfusions without carbenoxolone, but with a plateau after two hours (Fig 2 right graph).

## Tryptophan and serotonin

Tryptophan showed a steady increase in the fetal side throughout the 6 hours perfusion without reaching an equilibrium. At end of perfusion the concentration of tryptophan in the fetal system had reached approximately 70% of the maternal start tryptophan concentration. The tryptophan concentration in the maternal system fell to a plateau around 75% of initial concentration after 3 hours perfusion. When adding fluoxetine or PCPA there was a slight increase in the tryptophan concentration on the maternal side at start perfusion before falling to 75% of initial concentration, and with fluoxetine in the maternal system the tryptophan concentration seemed to increase after 2 hours perfusion reaching 90% of initial concentration after 6 hours perfusion (Fig 3 left graph).

Serotonin concentration declined rapidly in the maternal system and reached a plateau after 2–4 hours perfusion at 0.2% of initial serotonin concentration. In the fetal system a small flat peak was seen at 1 hour perfusion of 0.2% of initial maternal concentration, where after the concentration of serotonin in the fetal system fluctuated close to 0 for the remainder of the 6 hours perfusion. The addition of fluoxetine slowed the rapid decline in the maternal concentration of serotonin, which reached 6% of initial concentration in maternal system after 2 hours perfusion and further declined to 0.8% at 6 hours perfusion. In the fetal system the initial peak of fluoxetine at 1 hour perfusion was markedly increased to 1% of initial maternal concentration when adding fluoxetine (Fig 3 right graph).

The concentration of 5HIAA produced by the placenta reached a plateau after 120 minutes perfusion between 2.5–5% of the added amount of serotonin in maternal perfusate, and around 1% in the fetal perfusate with a second increase up to around 2% after 240 minutes perfusion. The addition of PCPA seems to slighty lower the production of 5HIAA (Fig 4).

Maternal stress related data (life events, Depression, Anxiety and Stress, PRA, neuroticism, conscientiousness and AFCE) and selected fetal exposure variables from the placental perfusions (cortisol concentration in fetal system at 30 minutes, difference in fetal system cortisol concentration between 30 and 300 minutes, serotonin concentration in fetal system at 60 minutes perfusion, difference in fetal system serotonin from 60 to 300 minutes, and placental production of 5-HIAA) are presented in S3 Table. The majority of the participating mothers had a normal depression, anxiety and stress levels, and normal or mild pregnancy related anxiety. Correlations test of stress parameters and data representing placental transport of cortisol and serotonin only showed correlations between related stress parameters, and correlations between related placental transport variables (S4 Table).

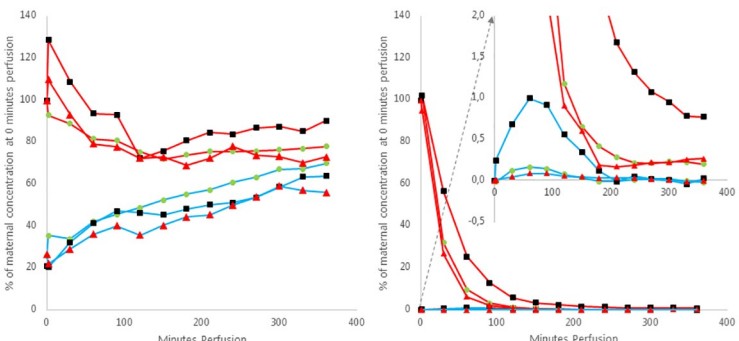

**Fig 3. Placental perfusion with tryptophan and serotonin with and without fluoxetine and PCPA.** The graphs shows maternal (red) and fetal (blue) mean concentration of tryptophan (left graph) and serotonin (right graph), without (green circle, n = 8) and with the addition of fluoxetine (black square, n = 4) and PCPA (red triangle, n = 5). The concentration is presented as percentage (%) of the concentration found in the maternal sample at time 0 minutes perfusion. The concentration of serotonin in all fetal system samples is subtracted the concentration of serotonin in the fetal system at time 0 minutes perfusion.

## Discussion

### Placental transport and metabolism

The endogenous concentration of cortisol, cortisone, tryptophan and serotonin fluctuate differently, and physiological levels in plasma during pregnancy is admittedly not representative for these fluctuations, especially at a cellular level. We aimed to use physiological start

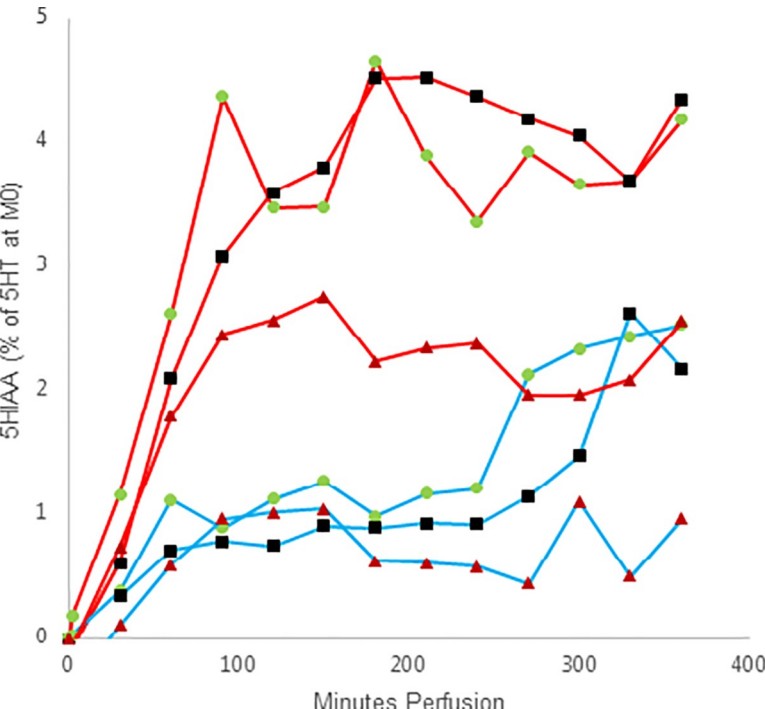

**Fig 4. Placental perfusion with 5-HIAA.** The graphs shows maternal (red) and fetal (blue) mean 5-HIAA concentration presented as percentage (%) of the 5-HT concentration in the maternal sample at time 0 minutes perfusion, without (green circle, n = 6) and with the addition of fluoxetine (black square, n = 4) and PCPA (red triangle, n = 5). The concentration of 5-HIAA in nM at each time point is subtracted the 0/before-sample concentration in nM in each perfusion on maternal and fetal side respectively.

concentrations of the four hormones, except for that cortisone was added at the same concentration as cortisol (10x physiological), and medical indicated doses for carbonoxolone, fluoxetine and PCPA.

The performed placental perfusions showed a rapid metabolism of cortisol into cortisone with 2% cortisol left in both fetal and maternal circulations. The rapid metabolism of cortisol into cortisone by placental 11β-HSD2 confirms previous placental perfusion studies of cortisol [14]. The addition of the enzyme inhibitor carbenoxolone affected this metabolism in our study, as the initial increase of cortisone in maternal circulation was not found in the perfusions with added carbenoxolone, and the concentrations in both fetal and maternal circulations were generally higher for cortisol and lower for cortisone. We still saw some metabolism of cortisol into cortisone when adding carbenoxolone probably caused by the relatively low concentration of inhibitor added.

In another study it is shown that the passive transport of cortisol in the placenta from maternal to fetal side is low at low concentrations (20–200 nM), but is increased at high (800 nM) concentrations, were 1/8 of the concentration (100 nM) is transferred to fetal side. When carbenoxolone is added the transfer is increased to 200 nM on fetal side [15]. Those data are the result of an open circuit placental perfusion, with a constant concentration of cortisol on maternal side, as opposed to the closed loop study in this paper were cortisol is metabolized and eliminated from the system on both fetal and maternal side. In the open-loop study it is also concluded that the metabolism of cortisone into cortisol is negligible (3–4 nM in the high concentration), although also inhibited by carbenoxolone. The placenta favoring the metabolism of cortisol into cortisone was confirmed by another closed loop study, which also demonstrated increased output of corticotropin-releasing hormone (CRH) into the fetal vein when perfusing the human placenta with cortisol [16].

Our perfusion studies of tryptophan and serotonin showed rapid decline and low levels at the end of the 6 hours perfusion of serotonin on both maternal and fetal side. The surplus concentration of tryptophan at end perfusion is probably endogenous tryptophan washed out of the placental tissue during the 6 hours perfusion. When adding fluoxetine to the perfusions there was an initial peak of serotonin concentration in the fetal system. The fetal supply of serotonin has been suggested to originate from tryptophan from the maternal blood metabolized into serotonin in the placenta by THP-1 [7], or more recently, to come from a pathway of serotonin from maternal blood through the placenta via SERT, gap junctions, organic cation transporters and P-glycoprotein [17]. In both these pathways the enzyme MAOA rapidly oxidizes and eliminates excess serotonin in the placenta. We showed increased initial concentration of both tryptophan and relative serotonin concentration in the maternal system when adding fluoxetine, so both these pathways could supply the initial peak of serotonin in the fetal circulation. Alternatively the increased serotonin could be a result of fluoxetine inhibiting uptake in platelets via SERT [18,19], although the fetal and maternal blood residues are mostly washed out from the placental lobule during the initial washing phase and pre-perfusion. The 5-HIAA levels we saw in the 6 hours placenta perfusion experiment only reached 7% of added serotonin, combined maternal and fetal system, which is less than expected.

When studying the placental transport of hormones there are possible interactions to take into account, in this study especially the feedback mechanism of cortisol on the metabolism of tryptophan into serotonin. Upregulation of TDO by cortisol causes increased metabolism of tryptophan into kynerunine and less into serotonin. The addition of cortisol in the perfusion studies of tryptophan metabolism is therefore physiologically relevant and a strength of this study.

The feed-forward mechanism of serotonin on cortisol cannot be studied on the isolated placenta, as this involves the maternal HPA axis [6]. However, it is interesting to consider the

delayed decrease in maternal serotonin concentration due to fluoxetine, and the possibility of this affecting the maternal HPA axis.

## Population and method

Our study population is a relatively unselected section of the population, and therefore represents pregnant women subjected to everyday stressors and pregnancy anxiety. We were not able to demonstrate a possible influence of self-reported maternal stress levels on placental tissue in this study, due to very similar normal stress levels in the participating women.

The chosen concentrations of hormones and enzyme/receptor inhibitors were based on the physiological levels, the limit of quantification of the substance quantification method, and the water-solubility of the substances.

The human *ex vivo* placenta perfusion technique is a superior alternative to animal studies. In the field of hormones and stressors during pregnancy the complexity of the human experience is difficult to replicate in an animal model, as the effect of pregnancy on the maternal physiology and mental state is unique. Moreover the developmental stages of the fetus are very species specific, but most importantly there are huge species differences in the placenta, which is the most diverse mammalian organ [20].

## Conclusion

Perfusions were performed on twenty-two placentas studying the effect of maternal stress levels and relevant enzymes on the placental transfer and metabolism on four chosen hormones: cortisol, cortisone, tryptophan and serotonin. The addition of carbenoxolone inhibitis the metabolism of cortisol in the placenta, and this increased the fetal levels of cortisol in our experiments as expected. Unexpectedly we saw a peak in fetal levels of serotonin an hour after beginning perfusion, in placental perfusions of serotonin with added fluoxetine that inhibits the serotonin transporter. This was observed together with a delayed decrease in serotonin concentration on maternal side which suggests that the increased fetal levels of serotonin is caused by increased serotonin concentration on the maternal side of the placenta, perhaps enabling other transplacental serotonin transport routes than through the SERT. The effect of maternal stress on placental transporters and enzymes represented in this study by adding inhibitors, can thus affect the fetal exposure to maternal cortisol and serotonin levels. The more direct interpretation is increased transient exposure to both cortisol and serotonin when treating pregnant women with enzyme inhibitors. The increased fetal exposure to serotonin when inhibiting the SERT is an interesting finding, especially as regards to pregnant women needing treatment with SSRIs due to major depressive disorder and PTSD. No effect on placental kinetics was seen on maternal stress levels during the pregnancy, which is probably due to small group size in this study and the low to normal stress levels in the majority of most participating mothers.

## Supporting information

**S1 Table. Successful perfusions in the maternal stress and placental function project according to success criteria.** The three success criteria are defined at different stages of the placental perfusion process: Criteria 1 is a successful cannulation, criteria 2 is a successful pre-perfusion, and criteria 3 is a successful 6 hour perfusion with added test-substances. (DOCX)

**S2 Table. Descriptive data.** Lifestyle, self-reported health and birth-related outcomes from the women donating the perfused placentas (n = 22).
(DOCX)

**S3 Table. Stress related questionnaire data from the women donating the perfused placentas, and selected fetal perfusion medium cortisol and serotonin variables.** State stress represented by DASS, PRA, life events and adjusted fetal cortisol exposure (AFCE), trait stress represented by NEO-FFI categories neuroticism and conscientiousness. Cortisol concentration in the fetal system after 30 minutes and from 30 to 300 minutes perfusion represents the rapid initial transfer of cortisol and the steady flow of cortisol through the placenta respectively. Serotonin concentration in the fetal system after 60 minutes and from 60 to 300 minutes perfusion represents the size of the initial peak in serotonin concentration in the fetal system. The concentration values in fetal system are presented as percentage (%) of the added cortisol and serotonin in maternal system (M0 sample). Sum of 5-HIAA concentration in fetal and maternal systems at 6 hours perfusion, presented as % of serotonin in M0 sample.
(DOCX)

**S4 Table. Correlations between maternal stress exposure and fetal exposure to tested hormones.** Bivariate correlations between: State stress represented by DASS, PRA, life events and adjusted fetal cortisol exposure (AFCE), trait stress represented by NEO-FFI categories neuroticism and conscientiousness. Cortisol concentration in the fetal system after 30 minutes and from 30 to 300 minutes perfusion represents the rapid initial transfer of cortisol and the steady flow of cortisol through the placenta respectively. Serotonin concentration in the fetal system after 60 minutes and from 60 to 300 minutes perfusion represents the size of the initial peak in serotonin concentration in the fetal system. The concentration values in fetal system are presented as % of the added cortisol and serotonin in maternal system (M0 sample). Sum of 5-HIAA concentration in fetal and maternal systems at 6 hours perfusion, presented as % of serotonin in M0 sample. Correlations were seen between stress variables: life-events, DASS depression, anxiety and stress, and FFI neuroticism. Reverse correlations were seen between related hormone exposure variables: fetal cortisol 30 minutes and Δ fetal cortisol 30–300 minutes, and fetal serotonin 60 minutes and Δ fetal serotonin 60–300 minutes.
(DOCX)

**S1 Data. Raw data from perfusions with added cortisol, cortisone, tryptophan and serotonin.** Data are presented in nM for cortisol, cortisone, tryptophan, serotonin and the metabolite 5-HIAA for each time-point in the perfusions from maternal and fetal system (n = 22). Missing data are due to analysis errors.
(XLSX)

## Acknowledgments

This project would never have succeeded without the invaluable assistance of Dr. Morten Hedegaard and the personnel at the maternity ward at Copenhagen University Hospital and all the participating pregnant women and their families. Sampling assistance was given by Julie Hansen Lawrence.

## Author Contributions

**Conceptualization:** Line Mathiesen, Lisbeth E. Knudsen.

**Formal analysis:** Line Mathiesen, Cecilie Bay-Richter.

**Funding acquisition:** Lisbeth E. Knudsen.

**Methodology:** Cecilie Bay-Richter, Nico Liebenberg.

**Project administration:** Line Mathiesen.

**Resources:** Cecilie Bay-Richter, Gregers Wegener.

**Supervision:** Lisbeth E. Knudsen.

**Visualization:** Line Mathiesen.

**Writing – original draft:** Line Mathiesen.

**Writing – review & editing:** Line Mathiesen, Cecilie Bay-Richter, Gregers Wegener, Nico Liebenberg, Lisbeth E. Knudsen.

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
