## [Decision Letter · Decision Letter 0]

8 Apr 2020

PONE-D-20-01763

Maternal stress and placental function; ex vivo placental perfusion studying cortisol, cortisone, tryptophan and serotonin.

PLOS ONE

Dear Mathiesen,

Thank you for submitting your manuscript to PLOS ONE. After careful consideration, we feel that it has merit but does not fully meet PLOS ONE’s publication criteria as it currently stands. Therefore, we invite you to submit a revised version of the manuscript that addresses ALL the points raised during the review process.

We would appreciate receiving your revised manuscript by May 23 2020 11:59PM. To enhance the reproducibility of your results, we recommend that if applicable you deposit your laboratory protocols in protocols.io, where a protocol can be assigned its own identifier (DOI) such that it can be cited independently in the future. For instructions see: http://journals.plos.org/plosone/s/submission-guidelines#loc-laboratory-protocols

We look forward to receiving your revised manuscript.

Kind regards,

Frank T. Spradley

Academic Editor

PLOS ONE

Journal Requirements:

2. Thank you for stating the following in the ***Financial Disclosure*** section:

"GW reports to have received lecture/consultancy fees/research support from H. Lundbeck A/S, Servier SA, Astra Zeneca AB, Eli Lilly A/S, Sun Pharma Pty Ltd, Pfizer Inc, Shire A/S, HB Pharma A/S, Arla Foods A.m.b.A., Alkermes Inc, Mundipharma International Ltd., and Jannsen Pharmaceuticals A/S. No other conflicts of interests are reported."

3. We noted in your submission details that a portion of your manuscript may have been presented or published elsewhere.

"Yes, the individual AFCE ratios and the Stress levels of the participating women are published as a part of a another paper as the participants in this paper are a subset of a previously studied population. The current studied subset is unidentifiable from the previous publication with is referenced in the text, and the study presented here is completely different. "

Reviewers' comments:

Reviewer's Responses to Questions

**Comments to the Author**

1. Is the manuscript technically sound, and do the data support the conclusions?

Reviewer #1: Yes

Reviewer #2: Partly

2. Has the statistical analysis been performed appropriately and rigorously? 

Reviewer #1: Yes

Reviewer #2: I Don't Know

3. Have the authors made all data underlying the findings in their manuscript fully available?

Reviewer #1: Yes

Reviewer #2: Yes

4. Is the manuscript presented in an intelligible fashion and written in standard English?

Reviewer #1: Yes

Reviewer #2: No

5. Review Comments to the Author

Reviewer #1: An innovative and valuable way to test the validity of the results in placenta-fetal metabolism in laboratory animals has been devised and used to study long term consequences of the gestational milieu. In this case it tests the impact of human maternal stress on the key modulators of the HPA axis in the fetus.

Reviewer #2: REVIEW PONE

Investigators collected term placentas from 22 women, and performed ex vivo perfusion concurrently (simultaneously) with cortisol, 5HT and tryptophan with and without inhibitors for treatments for 6 h successfully on 16 placentas. Cortisol was converted to cortisone by placenta, and was somewhat inhibited by carb, resulting in increased cortisol on fetal side. PCBA had no effect, whereas fluoxotine increased fetal side 5HT.

There is some merit in the manuscript, but the quality of writing is OK at best, and sloppy with an overuse abbreviations making it impossible to read fluidly and understand. I am not sure any of the maternal ‘stress’ information adds to the paper, as the sample size was too small to do analysis, and so mentioning it detracts from findings, etc., you may be better off indicating that placentas were collected as part of a larger study that was focused on understanding role of maternal stress on offspring development. There are no indicators of statistical significance of changes in hormone levels in response to treatments, that that because there are none? Please state within files.

line 17 which can cause changed health conditions, what is meant by this? I think you mean ‘has potential long-term effects on offspring’s development and health’

line 31 maternal 'stress' the n was really too small to analyze this

line 36, edit seen to 'evident'

needs to be define, 11BHSD2, line 59

line 67, as written suggests that 5HT is transported across placenta by SERT, whereas uptake of 5HT by SERT targets it for degradation via the pathway referred.

https://www.ncbi.nlm.nih.gov/pmc/articles/PMC3084180/ fetal 5HT source is not of maternal origin. Using additional genetic strategies, a new technology for studying placental biology ex vivo, and direct manipulation of placental neosynthesis, we investigated the nature of this exogenous source and uncovered a placental 5-HT synthetic pathway from a maternal tryptophan precursor, in both mice and humans.

although this paper suggests SERT SNP, may affect fetal exposure levels. https://www.ncbi.nlm.nih.gov/pubmed/27550733

ine 103, twenty-two

line 103-105, sentence needs editing for sense

line 127, define HAS

line 142, cortisole=cortisol

line 146, space between test

line 208 what was the number?

Line 226, define tendency, what was p-value, what was difference?

Line 232, the graphs show

line 241 at the end of

line 264 subtract ‘from’ the

line 340 write out TDo, over use of abbreviations

line 349, what is normal?

line 354, write out LOQ

As written the conclusion is confusing and not easy to follow

line 366, an hour after beginning perfusion of fluoxetine.

6. PLOS authors have the option to publish the peer review history of their article (what does this mean?). If published, this will include your full peer review and any attached files.

Reviewer #1: Yes: Elise P. Gomez-Sanchez

Reviewer #2: Yes: Theresa M Casey

---

## [Author Response · Author response to Decision Letter 0]

29 Apr 2020

Reviewers' comments:

4. Is the manuscript presented in an intelligible fashion and written in standard English?

Reviewer #1: Yes

Reviewer #2: No

Corrections are made as suggested by the reviewer

Reviewer #1: An innovative and valuable way to test the validity of the results in placenta-fetal metabolism in laboratory animals has been devised and used to study long term consequences of the gestational milieu. In this case it tests the impact of human maternal stress on the key modulators of the HPA axis in the fetus.

We thank the reviewer for this comment, we also believe the data presented are valuable in its’ field.

Reviewer #2: REVIEW PONE

Investigators collected term placentas from 22 women, and performed ex vivo perfusion concurrently (simultaneously) with cortisol, 5HT and tryptophan with and without inhibitors for treatments for 6 h successfully on 16 placentas. Cortisol was converted to cortisone by placenta, and was somewhat inhibited by carb, resulting in increased cortisol on fetal side. PCBA had no effect, whereas fluoxotine increased fetal side 5HT.

There is some merit in the manuscript, but the quality of writing is OK at best, and sloppy with an overuse abbreviations making it impossible to read fluidly and understand.

We thank the reviewer for this comment, and have tried to improve the quality of writing and avoid the overuse of abbreviations. 

I am not sure any of the maternal ‘stress’ information adds to the paper, as the sample size was too small to do analysis, and so mentioning it detracts from findings, etc., you may be better off indicating that placentas were collected as part of a larger study that was focused on understanding role of maternal stress on offspring development. 

We thank the reviewer for this comment. We believe the presentation of the maternal stress –though not showing significant result, could demonstrate the methods used and inspire future researchers in this area, if only to show that future research needs to include a larger population sample or more variation in population stress levels.

There are no indicators of statistical significance of changes in hormone levels in response to treatments, that that because there are none? Please state within files.

The statistical significance of treatments is not tested due the few repetitions, only observations are made. This is now stated in the manuscript.

line 17 which can cause changed health conditions, what is meant by this? I think you mean ‘has potential long-term effects on offspring’s development and health’

The change is made as suggested.

line 31 maternal 'stress' the n was really too small to analyze this

The statement “with no effect of adding PCPA” is deleted.

line 36, edit seen to 'evident' needs to be define, 

The change is made as suggested.

11BHSD2, line 59

The name of the enzyme is written in full at the first mention as suggested.

line 67, as written suggests that 5HT is transported across placenta by SERT, whereas uptake of 5HT by SERT 

targets it for degradation via the pathway referred.

https://www.ncbi.nlm.nih.gov/pmc/articles/PMC3084180/ fetal 5HT source is not of maternal origin. Using additional genetic strategies, a new technology for studying placental biology ex vivo, and direct manipulation of placental neosynthesis, we investigated the nature of this exogenous source and uncovered a placental 5-HT synthetic pathway from a maternal tryptophan precursor, in both mice and humans.

although this paper suggests SERT SNP, may affect fetal exposure levels. https://www.ncbi.nlm.nih.gov/pubmed/27550733

Thank you for this comment. We realize that the literature differs in the origin of the fetal serotonin. This manuscript presents data that shows the possible effect of increased levels of maternal serotonin causing a transient increased level of serotonin on the fetal side of placenta, thus supporting some correlation between maternal and fetal serotonin levels. If this increase is caused by transport via the SERT remains to be established.

ine 103, twenty-two

The correction is made as suggested.

line 103-105, sentence needs editing for sense

The sentence is edited as follows: Of these, 2 perfusions had a leak of medium from fetal circulation and were unsuccessful, 5 perfusions showed a leak of medium from the fetal circulation after 4 hours perfusion and were thus unsuccessful after this time-point, and 15 were successful 6-hour perfusions.

line 127, define HAS

HSA is written out as human serum albumin.

line 142, cortisole=cortisol

The change is made as suggested.

line 146, space between test

The change is made as suggested.

line 208 what was the number?

The number is added (n=22).

Line 226, define tendency, what was p-value, what was difference?

The statement is an observation and does not include statistical measures, the term “tendency” is deleted.

Line 232, the graphs show

The change is made as suggested.

line 241 at the end of

The change is made as suggested.

line 264 subtract ‘from’ the

The change is made as suggested.

line 340 write out TDo, over use of abbreviations

The change is made as suggested.

line 349, what is normal?

The term normal means relatively unselected in this case, and is therefore deleted.

line 354, write out LOQ

The change is made as suggested.

As written the conclusion is confusing and not easy to follow

Thank you for this observation, the conclusion is rewritten in part to aid understanding:

Perfusions were performed on twenty-two placentas studying the effect of maternal stress levels and relevant enzymes on the placental transfer and metabolism on four chosen hormones: cortisol, cortisone, tryptophan and serotonin. The addition of carbenoxolone inhibitis the metabolism of cortisol in the placenta, and this increased the fetal levels of cortisol in our experiments as expected. Unexpectedly we saw a peak in fetal levels of serotonin an hour after beginning perfusion, in placental perfusions of serotonin with added fluoxetine that inhibits the serotonin transporter. This was observed together with a delayed decrease in serotonin concentration on maternal side which suggests that the increased fetal levels of serotonin is caused by increased serotonin concentration on the maternal side of the placenta, perhaps enabling other transplacental serotonin transport routes than through the SERT. The effect of maternal stress on placental transporters and enzymes, represented in this study by adding inhibitors, can thus affect the fetal exposure to maternal cortisol and serotonin levels. The more direct interpretation is increased transient exposure to both cortisol and serotonin when treating pregnant women with enzyme inhibitors. The increased fetal exposure to serotonin when inhibiting the SERT is an interesting finding, especially as regards to pregnant women needing treatment with SSRIs due to major depressive disorder and PTSD. No effect on placental kinetics was seen on maternal stress levels during the pregnancy, which is probably due to small group size in this study and the low to normal stress levels in the majority of most participating mothers.

line 366, an hour after beginning perfusion of fluoxetine. 

The sentence is changed as follows: an hour after beginning perfusion, in placental perfusions of serotonin with added fluoxetine...

---

## [Decision Letter · Decision Letter 1]

18 May 2020

Maternal stress and placental function; ex vivo placental perfusion studying cortisol, cortisone, tryptophan and serotonin.

PONE-D-20-01763R1

Dear Dr. Mathiesen,

We are pleased to inform you that your manuscript has been judged scientifically suitable for publication and will be formally accepted for publication once it complies with all outstanding technical requirements.

With kind regards,

Frank T. Spradley

Academic Editor

PLOS ONE

Reviewers' comments:

Reviewer's Responses to Questions

**Comments to the Author**

1. If the authors have adequately addressed your comments raised in a previous round of review and you feel that this manuscript is now acceptable for publication, you may indicate that here to bypass the “Comments to the Author” section, enter your conflict of interest statement in the “Confidential to Editor” section, and submit your "Accept" recommendation.

Reviewer #2: All comments have been addressed

2. Is the manuscript technically sound, and do the data support the conclusions?

Reviewer #2: Yes

3. Has the statistical analysis been performed appropriately and rigorously? 

Reviewer #2: N/A

4. Have the authors made all data underlying the findings in their manuscript fully available?

Reviewer #2: Yes

5. Is the manuscript presented in an intelligible fashion and written in standard English?

Reviewer #2: Yes

6. Review Comments to the Author

Reviewer #2: The authors addressed my comments--but I suggest a very thorough read theough and editing by a native English speaker, there are still problems here and PLoSOne does not copy edit.

For example line 202 'are' shuld be were--as this was done in past/....authors use 'are' are' are, and then switch tense. Again the writing is quite sloppy and not a good reflection of qulaity.

7. PLOS authors have the option to publish the peer review history of their article (what does this mean?). If published, this will include your full peer review and any attached files.

Reviewer #2: No

---

## [Editor Report · Acceptance letter]

20 May 2020

PONE-D-20-01763R1 

Maternal stress and placental function; ex vivo placental perfusion studying cortisol, cortisone, tryptophan and serotonin. 

Dear Dr. Mathiesen:

I am pleased to inform you that your manuscript has been deemed suitable for publication in PLOS ONE. Congratulations! Your manuscript is now with our production department. 

With kind regards,

on behalf of

Dr. Frank T. Spradley 

Academic Editor

PLOS ONE